# Fiber Laser Alloying of Additively Manufactured 18Ni-300 Maraging Steel Part Surface: Effect of Processing Parameters on the Formation of Alloyed Surface Layer and Its Properties

**DOI:** 10.3390/ma16134732

**Published:** 2023-06-30

**Authors:** Jelena Škamat, Kęstutis Bučelis, Olegas Černašėjus, Simonas Indrišiūnas

**Affiliations:** 1Laboratory of Composite Materials, Vilnius Gediminas Technical University, Linkmenu Str. 28, LT-08217 Vilnius, Lithuania; 2Department of Mechanics and Materials Engineering, Vilnius Gediminas Technical University, Plytines Str. 25, LT-10105 Vilnius, Lithuania; kestutis.bucelis@vilniustech.lt (K.B.); olegas.cernasejus@vilniustech.lt (O.Č.); 3Department of Laser Technologies, Center for Physical Sciences and Technology, Savanoriu pr. 231, LT-02300 Vilnius, Lithuania; simonas.indrisiunas@ftmc.lt

**Keywords:** laser alloying, laser boronizing, surface hardening, laser powder bed fusion (LPBF), additive manufacturing, iron boride, hardness, wear resistance

## Abstract

The development of new efficient, economical, and safe methods for strengthening the working surfaces of parts is an important task in the field of improving the reliability and resourcefulness of critical equipment and structures. In the present paper, laser boronizing is investigated as an alternative method for improving the wear resistance of maraging steel parts manufactured by laser powder bed fusion (LPBF). After LPBF, the specimens’ surface was covered with an amorphous boron paste (0.03–0.6 mm) and laser processed with a continuous-wave fiber laser in melting mode (λ—1070 nm; power—300 W; spot Ø—1.0 mm) at 500–1500 mm/min laser beam scanning speeds. Scanning electron microscopy, X-ray microanalysis, Knoop hardness, and dry sliding wear tests were applied to investigate the geometry, microstructure, hardness and its distribution, heat-affected zones, wear resistance, and wear mechanism of the alloyed layers. The boronized layers of thickness ~280–520 µm with microstructure from hypoeutectic to borides’ mixture were obtained, whose hardness varied from ~490 to ~2200 HK0.2. With laser boronizing, the wear resistance was improved up to ~7.5 times as compared with aged LPBF samples. In further method development, the problem of thermal cracking and softening of the heat-affected zone should be solved.

## 1. Introduction

Additive manufacturing (AM) is the officially established name for technologies that form 3D parts from digital models by adding materials (plastic, metals, etc.) layer-by-layer [1]. AM processes, which were first created for rapid prototyping (RP), have now reached a high enough development level for the manufacturing of finished products. The laser powder bed fusion (LPBF) process is one of the main representative processes developed for metal part production by AM. During LPBF, a laser beam selectively fuses powdered material by scanning cross sections generated from a 3D digital description of the part. During the build cycle, the platform on which the build is repositioned lowers by the thickness of one layer; the process continues until the build or model is completed [2]. Due to the possibility to produce parts with complex and/or personalized internal and external geometries along with other benefits of AM, LPBF attracts growing interest in various fields of engineering, including medical [3], aerospace and aviation [4,5], bridges [6], and tooling [7]. During the LPBF process, volume micro-elements of material experience rapid heating above their melting point and rapid solidification; moreover, material of the already synthesized volume is being reheated numerous times when further fusing of neighboring tracks and following layers is performed. Therefore, alloys sensitive to thermal cracking are not usable, and mostly carbon-free steels have been used by now [8]. Maraging steel (MSt), having excellent weldability and high resistance to thermal fatigue due to the lack of carbon, is one of such materials that showed good suitability for the LPBF process [9,10,11,12,13,14]. According to the reported results, the mechanical properties of MSt parts produced by LPBF at carefully optimized parameters are not inferior to those typical for wrought products, which opens new perspectives in the fields where MSt is traditionally used—first of all, in aerospace and tooling.

Despite the high strength of MSt (~2000 MPa and over), it has only moderate hardness, which necessitates hardening the surface of the MSt parts for exploitation under friction and wear conditions. The main method used to strengthen the surfaces of MSt is nitriding, as a result of which a hard, wear-resistant nitride-containing layer is formed on the surface. The main disadvantages of nitriding are the long process duration, heat treatment stress, high cost, and environmental impact due to nitrogen oxide emissions during processing cycles. Therefore, it is an urgent task to find new solutions and develop new methods for strengthening steel surfaces based on the use of more economical and environmentally friendly processes. Laser processing is considered a technology that makes manufacturing more sustainable [15]. The known advantages of laser processing, such as versatility, flexibility, high controllability of process parameters, and a high degree of process automation, against the background of the rapidly developing laser industry, which provides a reduction in prices for laser equipment and increases its availability, make laser processing a very promising method for increasing the wear resistance of important parts, especially for AM production, where the integration of laser-based processes would be more simple. By changing the energy parameters of the laser beam and the scanning speed, it is possible to process metal materials in heating, melting, or ablation modes, implementing various processing technologies, one of which is laser alloying. This method allows changing the chemical and phase composition of the surface of the material by melting the surface with a laser beam in the presence of alloying additives preplaced on it. In terms of wear resistance improvement, alloying additives, which increase the hardness, are effective—hard compounds (oxides, carbides, nitrides, borides) or elements that form such compounds in reactions with metals.

From the general theory of the thermochemical treatment of steels, it is known that, in terms of improving wear resistance, the most effective method is boriding, which is due to the high hardness of iron borides (17,000–22,000 MPa for FeB and 14,000–18,000 MPa for Fe_2_B [16]). Despite the fact that thermochemical boriding has been known since the early 1910s, the interest of scientists in this technology did not subside, as confirmed by the publications of recent years [17,18]. However, conventional thermochemical boriding is not suitable for maraging steels due to its very high process temperature, which has a detrimental thermal effect on the mechanical properties of maraging steels due to the formation of reverted austenite [19]. Laser alloying, which was not previously used for MSt, could be an alternative to thermochemical processes; it allows alloying surface layers with various compounds and elements but has a limited thermal effect on the product as a whole, which makes this method possible to apply to temperature-sensitive MSt. The effectiveness of the laser boronizing method on nickel alloy [Inconel 718] and steels [41Cr4, 100CrMnSi6-4, C20, C45, C90, EN25] is already confirmed by the studies presented in [20,21,22,23,24] and summarized in Table 1. According to the results reported, uniform, hard (up to ~1600 HV) iron-boride-containing layers may be obtained on steels, providing improved wear resistance. Differently from MSt, which contains only 0.03 wt.% C, the steels investigated in the mentioned works contain 0.35 to 1.03 wt.% of carbon; as a result, martensite presence was confirmed in all cases, which could contribute to hardness increase as well. Despite the martensite formation, in cases where hardness was below ~1400 HV, no cracks in the appearance were reported. The formation of cracks is mentioned in work [21], where harder boride layers are obtained (up to ~1600 HV), but this issue was not considered more closely.

In the previous work of the authors [25], the application of laser boronizing technology to modify the surface of additively manufactured MSt parts was demonstrated; single passes were formed in a wide laser parameter range. It was found that the laser melting of a surface with preplaced amorphous boron paste results in the solidification of the boride-containing layer, which has increased hardness (up to 1700 HK). Boron paste was preplaced on the side-on surface of LPBF samples using a developed surface morphology formed after LPBF for boron paste thickness control. Since MSt is sensitive to heating, processing was performed before heat treatment of LPBF the part and without preheating. The absence of preheating simplifies the technology but results in higher cooling speeds that contribute to thermal stresses and the risk of crack formation. As a result, the appearance of cracks was observed in a number of specimens. It was also determined that Fe_3_B-type borides dominate besides other boride phases identified in alloyed layers, which corresponds to the supercooled Fe-B alloy phase diagram and is associated with high cooling rates as well. Full thermal treatment (which included solution annealing at 840 ± 5 °C for 2 h, cooling in air, and aging at 490 ± 5 °C for 2 h) of laser-boronized parts resulted in boronized layer hardness decreasing by up to ~300 HK0.2 numbers, indicating that such a sequence of operations is not effective. Respectively, the present work, which continues the previously started experiments, investigates the laser boronizing of already aged parts. In this case, an important aspect of the study was the evaluation of the effect of heating the base metal in the heat-affected zone and the softening effect associated with the formation of reverted austenite. In addition, the effectiveness of preheating to prevent crack formation in boronized layers was evaluated, applying the maximum preheating temperature as high as the maximum working temperature for aged 18Ni-300 MSt (~400 °C). To better understand the effect of MSt alloying with boron at different boron concentrations, various boron paste thicknesses were applied, providing a boron-to-steel mixing ratio in a wider range.

The present paper reports the results of laser boronizing experiments performed on additively manufactured and aged maraging steel parts, which may contribute to the further development of new laser-based, efficient, economical, and safe methods for strengthening the working surfaces of critical parts.

## 2. Materials and Methods

### 2.1. Samples Material and Manufacturing

The samples for experiments were manufactured by the LPBF process using Concept Laser M3 equipment. Steel powder of grade DIN 1.2709, also known as 18Ni-300 maraging steel (chemical composition in wt.%: 0.03% C; <0.1% Si; <0.1% Mn; (17–19)% Ni; 4.8% Mo; <0.8% Ti; (8.5–9.5)% Co; <0.1% Al; Fe–balance), was used. This is one of the most widely used and commercially available maraging steel grades. The powder particle size declared by the powder manufacturer was between 7 and 30 μm. The main characteristics of the equipment and process parameters were as follows: laser wave length—1064 nm; laser power—100 W; laser spot size—0.2 mm; thickness of single layer—0.03 mm; sintering rate—0.2 mm/s; shielding gas—Ar 0.75 L/h. Square prism specimens with dimensions of 20 × 20 × 6 mm were produced, which were separated from the substrate (made of the same steel grade) by the electric discharge cutting method and cleaned in an ultrasonic bath in C_3_H_8_O solution for 15 min at 40 °C. Then, the samples were heat treated by solution annealing at 820 ± 10 °C, followed by air cooling to lath martensite and aging at 490 ± 5 °C.

### 2.2. Laser Boronizing Process

Fiber laser machine FANUCI^®^ PRO 1500 (Fanuci, Gdansk, Poland) with a single module optical fiber laser source with a wavelength of 1070 nm was used for surface laser alloying. The main processing parameters were as follows: continuous laser emitting; laser power—300 W; laser spot diameter—1 mm; laser operating speed between 500 and 1500 mm/min; shielding gas—argon (consumption—15 L/min). The efficiency of the laser was >25%, that is, the average laser beam power density was >9554 W/cm^2^ during the experiments, providing the processing in the melting mode. Before laser alloying, the surface of the samples was pre-ground to align the surface and provide micro-roughness for better paste fixing (*Ra* = 9.2 µm), washed in isopropyl, and dried, and then amorphous boron paste was preplaced on them. For the preparation of the paste, amorphous boron powder (96.7 wt.% B; ≤1.0 μm average particle size; very dark brown color, as declared by manufacturer Sigma-Aldrich) was mixed with 3 wt.% organic binder and then with alcohol to obtain fluid consistence. Coated samples were dried in an electric furnace at 150 ± 5 °C for 1 h. The laser alloying experiments were conducted applying the parameters listed in Table 2 with samples being preheating to a temperature *T*_PH_ of 200 °C or 400 °C, as shown in Figure 1. In the first experiment stage, preheating to a temperature *T*_PH_ of 200 °C was carried out and the step between the adjusted tracks *s* was 0.7 mm. A few preliminary experiments have shown the appearance of cracks; therefore, *T*_PH_ was increased to 400 °C—this is the maximum possible temperature that does not cause undesirable structural changes in aged maraging steel. After the preheating temperature and thickness of the boron paste layer were increased, the geometry of individual molten pools changed; therefore, the step between adjacent tracks was reduced to 0.4 mm.

### 2.3. Characterization of Boronized Surfaces

For the microscopical analysis of the laser-alloyed surfaces and wear tracks, a scanning electron microscope SEM JEOL JSM-7600F (JEOL, Akishima, Japan) equipped with an energy dispersive spectrometer (EDS) Inca Energy 350 SDD X-Max 20 mm^2^ (Oxford Instruments, Oxford, UK) for X-ray microanalysis was used. SDD X-Max is a silicon drift detector (SDD) capable of analyzing all elements from Be to Pu (including boron). The main analysis parameters were: 10 kV accelerating voltage; ~8 mm working distance; mixed secondary and backscattered electron signal for imaging. For the microscopic analysis, the samples were sectioned, mounted, grounded, polished, and etched (H_2_O, CH₃COOH, HCl, and HNO₃ in a 1:1:4:1 ratio) using conventional techniques for metallographic analysis (the last polishing step was performed using 0.2 μm fumed silica suspension). The depth of the alloyed layers was measured from SEM images using SEM JEOL JSM-7600F software PC-SEM Ver2.1.0.9 for image analysis and measurements.

The hardness study was conducted using the microhardness tester Zwick Roell ZHμ (ZwickRoell GmbH & Co. KG, Ulm, Germany). Measurements were carried out using the Knoop method with a 200 g load and a 15 s duration on the mounted, ground, and polished cross-sections of the samples. The indentations were made across the molten pool depth and also parallel to layers in different zones of the molten pool. The average values of microhardness were calculated as an arithmetic mean of 15–20 individual measurements. The average hardness is presented with a standard deviation calculated as a STDEV.P function using Microsoft Excel 2016 software. The paper presents the microhardness distribution profiles across the depth of the alloyed layers obtained by making indentations at a depth of 50 μm from the surface with a step of 100 μm. The distribution of microhardness was analyzed in three to five zones of the alloyed layers for each sample. In the paper, for each tested sample, the most typical profile is given, which is a sequence of values obtained from individual measurements.

The tribological properties of the experimental alloyed surfaces were evaluated by the “ball-on-disc” dry sliding tests. The tests were performed using a Microtest tribometer. Before the test, the surface of the sample to be tested was pre-polished to *Ra* ~0.46 μm. The tribological testing parameters were as follows: sliding distance—200 m; sliding speed—200 rpm; radius of the trajectory—2.0 mm; load—20 N; temperature of the experiment—22 ± 1 °C; indenter—a 6 mm diameter ball made of tempered stainless steel AISI52100. The wear resistance of the coatings was evaluated by mass loss. The analytical balance Precisa XR 205SMDR with an accuracy of 0.00001 g was used to measure the mass of the samples before and after the tribology test. Before weighing, the test samples were cleaned in an ultrasonic bath for 15 min and dried in a hot air stream.

## 3. Results

### 3.1. Effect of Different Factors on the Formation of Laser-Alloyed Layers

#### 3.1.1. Molten Pool Geometry

Figure 2 shows laser-boronized layers obtained with samples preheated to 200 ± 5 °C and a 0.03 mm thickness of preplaced boron paste layer. The shallow and wide molten pools were obtained at the applied process parameters. The shape of the boundary between the individual molten pools and bulk material may be considered a semi-circle. With the increase in laser operating speed from 500 to 1500 mm/min, the molten pool depth decreased from ~412 to ~260 μm and the width from ~914 to ~586 μm; the dependence of both parameters on the laser speed is well described by linear regression: width = −0.3279*x* + 1069.7 (R^2^ = 0.9927) and depth = −0.1522*x* + 494.36 (R^2^ = 0.9799), where *x* is laser operating speed in mm/min. The width-to-depth ratio was quite stable with laser speed and ranged between 2.04 and 2.26.

The increase in preheating temperature to 400 °C led to a change in molten pool geometry (Figure 3a–c): deeper and narrower molten pools were obtained with a shape that can be described as a semi-oval. The average molten pool depth, which largely determines the mixing ratio of parent metal with alloying material, was increased by ~27% and ~28% at 500 and 1000 mm/min laser speeds (respectively) and by ~64% at 1500 mm/min. The width-to-depth ratio was reduced around two times and varied between ~1.01 and ~1.18. It was also observed that when the laser scanning speed reached 1500 mm/min, the shape of the molten pool changed from semi-oval to bowl (or vase).

In further experiments, the thickness of the preplaced boron paste layer was increased to 0.2 mm, 0.45 mm, and 0.6 mm. The general view of the alloyed layers obtained is presented in Figure 3d–l. In the case of 0.2 mm boron paste thickness, the expressed transition from molten pool shape to bowl (or vase) can be observed (Figure 3d–f). With thicker boron layers (0.45 and 0.6 mm), wider and shallower molten pools were formed (Figure 3g–l).

As was shown by Siao and Wen [26], molten pool shape is largely determined, among other things, by the Marangoni effect. The Marangoni effect is a mass transfer phenomenon when fluid from areas with low surface tension is transferred to areas with higher surface tension [27,28]. A negative value of *d*γ/*dT*, which indicates that the surface tension (γ) reduces with an increasing temperature, prompts a radial outward flow of the molten metal and hence produces a wider and shallower molten pool. A positive surface tension gradient leads to a radial inward flow and forms a deeper and narrower molten pool [29,30,31]. The surface tension gradient (*d*γ/*dT*) is both composition- and temperature-dependent [18]. Here, the increase in thickness of the boron paste layer resulted in an increase in the mixing ratio of boron with steel, leading to a change in the elemental and phase composition of the molten pool. This may be the reason for the *d*γ/*dT* change, leading to the variation in molten pool shape. Moreover, the gradual decrease in molten pool depth can be pointed out by raising laser operation speed and boron paste thickness (Figure 4). The effect of laser operation speed is caused by a reduction in heat input and is widely described in the literature. While the thickness of the preplaced boron paste layer, besides the Marangoni effect, influences the penetration depth in another way: the part of laser beam energy is absorbed by the preplaced boron paste layer; this may cause the molten pool to “shift” upward, reducing the depth of the resultant solidified molten pool. In the case of higher laser speeds (1000 and 1500 mm/min), corresponding to lower heat input, the drop in molten pool depth with boron paste thickness is quite well described by linear regression. At higher heat input (500 mm/min laser speed), the variation in pool depth has a more complicated character.

#### 3.1.2. Hardness of Laser-Boronized Layers

The average hardness of the obtained laser-boronized layers ranged in a wide interval—between ~490 and ~2224 HK0.2 (Figure 5a). The hardness of laser-boronized layers largely depends on the amount of boride phase formed, which is predetermined by the boron-to-parent steel mixing ratio. As was expected, the hardness increase was observed with both the increase in boron paste layer thickness and the raising of the laser operating speed, leading to the reduction in molten pool depth. Dividing the boron paste thickness *h* by the molten pool depth *H*, the parameter *h*/*H* was calculated, which determines the boron-to-parent steel mixing ratio at different boron paste thicknesses and laser operating speeds. Figure 5b shows the relationship between the calculated *h*/*H* values and the average hardness of the alloyed surface.

#### 3.1.3. Cracking of Laser-Boronized Layers and Surface Appearance

The appearance of cracks in thermochemically boronized layers is a known problem. The main reason indicated is the presence of a high-hardness boride phase (for steels—FeB), which is more brittle and sensitive to the formation of macro- and microcracks as compared with lower borides, for example Fe_2_B [32]. In general, the brittleness of boronized layers is controlled by a number of factors: the electron structure of phases formed, the correlation between formed boride phases in a layer, the morphology of borides, their stress state, the continuity of a boride layer on the processed surface, etc. [33]. The hardest boronized layers obtained in the present experiment also showed extensive surface chipping during cutting and further preparation of samples, which can be associated with the high brittleness of the layers (Figure 3h–l). In Figure 3f–h,j, in addition to surface chipping, vertical cracks can be observed, associated with thermal stresses. Figure 6 shows the surfaces of samples after laser boronizing; the typical appearance of cracks formed and surface brittle fracture can be observed from Figure 6d–f. The formation of a boronized layer during the laser alloying process differs from that during thermochemical treatment; the surface is locally heated by a laser beam, and an alloyed layer is formed “pass-by-pass” by solidification from the melt state. As is well known from the welding theory, non-uniform heating/melting of the surface causes the formation of residual tensile stresses (thermal stresses) because surrounding much colder material restrains free contraction of the solidifying and cooling molten pool; with further cooling and solidification, the thermal stresses increase. The structure of the laser-boronized layer differs from that of the thermochemically treated layer as well: in the laser-boronized layer, along with hard boride phases, a eutectic structure is also formed at lower boron concentrations. A eutectic phase crystallizes at a lower temperature, and more significant thermal stresses are formed. When the level of thermal stresses is high, the brittle phases cannot withstand the tension due to limited plasticity; if thermal stresses reach a high level while eutectic phases with a low melting point are not yet crystallized, the melt will not withstand the tension and will crack. Therefore, the presence of both eutectic and brittle phases facilitates the formation of cracks, while the main way to suppress crack appearance is a reduction in thermal stresses.

The temperature gradient has a significant effect on the stress level. The preheating of the part to be processed allows a reduction in the difference in temperatures between the molten pool and the colder rest of the part, which results in a reduction in thermal stresses and fewer or no cracks in the processed layers. The experiment showed that under these processing conditions, the preheating temperature of 400 degrees is not sufficient to fully prevent the appearance of thermal cracks in boronized layers.

As the elemental and phase composition of the surface is changed during the alloying process, the difference in thermal shrinkage coefficients between the base metal and the modified surface layer, as well as a change in specific volume due to the formation of boride phases, also may play an important role here. The density and thermal expansion coefficient α of 18Ni-300 MSt are 8.1 g/cm^3^ and 10.3 × 10^−6^ K^−1^, respectively. For tetragonal Fe_2_B boride, such values are indicated in the literature: 6.93–7.43 g/cm^3^ and 7.85 × 10^−6^ K^−1^. Since the specific volume of this boride is ~8–14% larger and α is ~24% less than that of MSt, the formation of this type of boride should not contribute to the formation of solidification cracks. The density of rhombic FeB boride is even smaller (6.47–6.8 g/cm^3^), while α is ~2.2 times greater (23 × 10^−6^ K^−1^) than for MSt. As a result, with the crystallization of a large amount of FeB, the specific volume increases, an even more uneven convex surface of the boronized layer is formed, in which, with further cooling, cracks develop due to significant shrinkage of the brittle layer. Accordingly, the formation of large amounts of higher borides is not desirable.

As can be observed from Figure 6a–c, the layers obtained at the thickness of boron paste of 0.03 and 0.2 mm have a fairly uniform and even surface, characterized by low micro-roughness and the absence of cracks and pores. Roughness parameters for sample series processed at boron paste thicknesses of 0.03 and 0.2 mm were as follows: *Ra* along the laser pass—from ~0.73 to ~1.48 µm; *Ra* across laser passes—from 1.35 to ~2.42 µm. To remove scaliness and provide the required surface quality, post-processing by grinding and polishing can be performed. The surface of the samples series laser boronized at boron paste thicknesses of 0.45 mm and 0.6 mm (500 mm/min laser speed), besides cracks, also showed increased irregularity and undulation of the surface (Figure 6d,e). The roughness parameters for these sample series could not be measured. The alignment of such surfaces would require the removal of thicker layers and the use of harder tools, given the quite high hardness of the surfaces. With a boron thickness of 0.6 mm (at 1000 and 1500 mm/min), it was not possible to form a continuous uniform layer—the breaking out of entire strips of the surface layer along the laser tracks can be observed in Figure 6f.

The laser boronizing method investigated in the present study is aimed first of all at improving the wear resistance of the MSt surface. However, the presence of a hard layer of limited plasticity on the surface may significantly affect the mechanical behavior of the part as a whole, especially when exploiting cyclic loads. As was shown in [21], the surface cracks formed during laser re-melting were the reason for a relatively quick first fatigue crack. Accordingly, the main task for the further development of this method is to solve the problem of crack formation. The following main ways can be noted: limiting the formation of a very large amount of very hard higher borides by adjusting the boron-to-steel mixing ratio; reducing the level of thermal stresses that may be reached by reducing the power applied (but providing molten pool temperature sufficient to melt all components), and reducing laser operating rates, providing lower cooling rates and slower solidification of the molten pool. Increasing the preheating temperature may be considered as well; however, in this case, the possibility of applying the aging of parts after boronizing should be evaluated.

#### 3.1.4. Microstructure of Laser-Boronized Layers

The microstructure of the obtained boronized layers varied from hypoeutectic (Figure 7a) containing a low amount of the boride phase to borides’ mixture (Figure 7d–f) and in general was in accordance with the Fe-B phase diagram; typical microstructures obtained at different boron paste layers and laser speeds are shown in Figure 7.

The layers obtained with 0.03 mm boron paste thickness (preheated to 200 °C and 400 °C) had only hypoeutectic microstructure (Figure 7a), with an increasing amount of eutectic phase at higher laser speeds. The individual hardness values of these layers varied from 487 to ~590 HK0.2. The “0.2 mm/500 mm/min” samples’ series also showed hypoeutectic microstructure, but with a higher eutectic amount and a local hardness reaching ~840 HK0.2. The microstructures of other samples were not so homogenous; each layer had several different zones. In “0.2 mm/1000–1500 mm/min” samples, the following microstructures were observed: hypoeutectic consisting of γ-Fe solid solution dendrites in eutectic matrix + eutectic (Figure 7b) + hypereutectic consisting of primary borides in eutectic matrix (Figure 7c)—local hardness was up to 1084 HK0.2. In samples “0.45 mm/500–1000 mm/min” and “0.6 mm/500 mm/min,” two types of microstructures were mainly observed: borides in eutectic matrix (as shown in Figure 7c) and microstructure shown in Figure 7d, which, according to X-ray microanalysis (results are shown in Figure 7e) and measured hardness (up to 1544 HK0.2), was identified as the mixture of two types of borides—most likely (Fe, Co, Ni, Mo)B (designated as MB) and (Fe, Ni, Co)_2_B (designated as M_2_B). The rest of the samples, which showed the highest hardness up to 2224 HK0.2, had a prevailing microstructure, as shown in Figure 7f—mixture of borides with dominance of higher MB borides.

### 3.2. Effect of Laser-Boronizing Parameters on HAZ

Very high yield strength (up to ~2420 MPa for commercial grades) of MSt is obtainable after two-step heat treatment, which includes solution annealing at ~820 °C followed by air cooling to lath martensite and aging at ~490 °C, which provides the precipitation of the dispersive intermetallides in a quite soft martensite matrix (~ 30 HRC). The influence of temperature on the mechanical properties of MSt is presented in [34]. When the temperature rises to ~400 °C, a gradual decrease in strength is observed, followed by a rapid drop at temperatures above ~450 °C, associated with the reversion of the martensite matrix to austenite. Therefore, MSt can be exploited for prolonged service at temperatures up to ~400 °C. The laser processing typically provides local heating and high cooling rates. However, the parent metal near the molten pool is also heated, and its temperature here ranges between the melting point (near the fusion line) and preheating temperature. In this heat-affected zone (HAZ), softened layers are formed, as Figure 8 shows. For all the samples investigated, two heat-affected zones were clearly distinguishable in the optical micrographs and some SEM images after etching. A narrower and clearer zone has formed near the fusion line (HAZ_1_). Further, a wider zone can also be seen (HAZ_2_), which differs little in etching from the base material but has a rather clearly visible border with it.

The depth profiles of hardness, presented in Figure 9a,b, revealed a significant softening effect near the fusion line (HAZ_1_), where the heating temperature is close to the melting point of the MSt (*T_m_* = 1413 °C). Here, metal was heated above austenization temperature and then cooled to martensite. The lowest hardness value of HAZ_1_ slightly differed for samples processed at differing parameters. For samples processed at lower laser speeds (500 and 1000 mm/min), the hardness of HAZ_1_ ranged between 329 and 369 HK0.2. These hardness values are comparable with those for non-aged lath martensite matrix. Samples processed at 1500 mm/min laser speed showed the hardness of HAZ_1_ ~429 HK0.2, which may be associated with incomplete austenization of this zone before cooling when lower heat input is applied.

HAZ_2_ was characterized by higher hardness and gradual hardness growth up to the hardness of aged MSt (slightly above 600 HK0.2). This part of HAZ was heated to a temperature between the austenization beginning and the preheating temperature applied during the experiment. The softening of this zone is associated with the formation of reverted austenite, whose amount increases with time at high temperatures and, consequently, increases with the heat input applied. The thickness of both the most softened HAZ_1_ and the whole HAZ is heat-input-dependent, which is demonstrated in Figure 9c,d: at a lower laser speed, corresponding to higher heat input, the thicker HAZ_1_ and HAZ are formed. The increase in preheating temperature also resulted in thicker HAZ_1_ and HAZ. The effect of preplaced boron paste thickness had a more complicated nature: firstly, when the thickness of the boron layer was increased from 0.03 to 0.2 mm, a reduction in HAZ was observed, and then, with further boron paste thickness increases, the HAZ showed slight growth as well. Possibly, besides heat input, other factors could influence the HAZ here: the change in molten pool shape, depth, and volume with boron paste thickness, and the change in cooling conditions due to the presence of boron paste on the sample top. In general, under actual experimental conditions, the thickness of the HAZ_1_ was ~40–130 μm, and the total HAZ was 200–650 μm thick.

### 3.3. Tribology of Laser-Boronized Surfaces

In the present laser alloying experiment, boronized surfaces in a wide hardness range from ~490 to ~2200 HK0.2 were obtained. Comparative two-body dry sliding wear tests were conducted to assess their wear resistance. From the series of samples obtained at 0.03 mm boron paste thickness, the one processed with a 200 °C preheating temperature was chosen for testing. In addition, studies were not carried out on samples with the highest hardness, which showed extensive surface chipping when preparing samples. The samples’ mass loss after wear tests is presented in Figure 10. As was expected, relatively soft control LPBF samples have shown more extensive wear. Microscopic analysis of the surface of the LPBF samples after tests revealed the presence of numerous traces, indicating the predominance of the adhesion mechanism of wear (Figure 11a). The wear rate was significantly (~2.6 times) reduced after aging, which is due to an increase in overall hardness due to the formation of dispersed intermetallides with increased hardness. A change in the structural-phase composition has also led to a change in the wear mechanism. Scratches and wear products (debris) were found on the tested surface (Figure 11b). The presence of hard intermetallides in the test material leads to more intense abrasion of the counter-body; the appearance of quite hard wear products and also intermetallic particles between the surfaces leads to a change in wear mechanism to an abrasive one. All boronized samples showed a decrease in wear, with the exception of a series of 0.03/500 samples having the lowest hardness ~ 490 HK0.2. For 0.03/500 samples, the wear mechanism was found to be similar to that of AG (Figure 11c), but mass loss was ~12% more. Despite the fact that the hardness of the samples 0.03/1000 and 0.03/1500 was less than that of the control samples, they showed an improvement in wear resistance of ~22–25%. The maximum hardness values obtained on these samples were close to the hardness of control samples (~600HK0.2). Thus, the presence of a harder (compared to intermetallides) boride strengthening phase in boronized layers, with a similar average hardness of these layers, provided higher wear resistance.

Moreover, on 0.03/1500 samples, traces of brittle cracking and delamination were also found (Figure 11d), which may be the result of surface plastic strengthening due to repeated deformation by the counter-body. All samples obtained at a boron paste thickness of 0.2–0.6 mm showed high wear resistance—on average, up to 7.5 times higher than the control-aged one. The mass loss was mainly due to the smoothing of the irregularities formed during the preparation (grinding) of the samples before the test. No mass loss was found in some tests. In addition, areas containing chromium were found on the surfaces of the samples after tests (Figure 11e), which indicates that the material of the counter-body adhered to the wear surface; that is, mostly the counter-body experiences wear. In the hardness range from ~700 to ~1450 HK0.2, there was an improvement in abrasion resistance with an increase in hardness. With a further increase in hardness, an increase in the mass loss of samples was established due to brittle fracture and chipping of the surface of samples at contact load by the counter-body (Figure 11f).

## 4. Conclusions

In the present study, the fiber laser surface alloying process of additively manufactured maraging steel parts was investigated. The following conclusions were drawn from the results of the experimental research described in this paper:Fiber laser processing of maraging steel surfaces, pre-coated with amorphous boron paste 0.03–0.6 mm thick and with a continuous laser emitting in melting mode at 300 W power and laser scanning speeds 500–1500 mm/min, allows boronized layers ~280–520 mm thick with hardness from ~490 to ~2200 HK0.2 to be obtained. The microstructure of the alloyed layers varied from hypoeutectic containing low amount of borides phase to mixture of borides (according to X-ray microanalysis: MB, where M is Fe, Co, Ni, and Mo in order of decreasing concentration, and M_2_B, where M is Fe, Ni, and Co in order of decreasing concentration), and in general was in accordance with the Fe-B phase diagram.The molten pool shape and depth changed with preheating temperature; the increase in preheating temperature resulted in deeper and narrower molten pools. With the increasing boron concentration (at thicker boron paste layers and faster laser speeds decreasing the molten pool depth), the shape of the molten pool was changed as well; in general, wider and shallower pools were formed.In the entire interval of the process parameters, a heat-affected zone HAZ is formed, consisting of two zones—a narrow (40–130 μm) most softened (~330–430 HK) zone located near the fusion line, and a wider zone (200–650 μm) with hardness increasing as it moves away from the fusion line from ~450 to ~600 HK0.2. The heat input had the greatest influence on the thickness of the zone.Two types of cracks were found on doped layers: brittle cracks and chips due to the formation of very hard higher borides, and thermal cracks due to residual tensile stresses, eutectic formation, limited plasticity of boride phases, and an increased coefficient of the thermal expansion of FeB boride. The preheating temperature of 400 degrees was not sufficient to avoid the formation of thermal cracks.All boronized samples showed a decrease in wear, with the exception of a series of samples having the lowest hardness at ~ 490 HK0.2. In the hardness range from ~700 to ~1450 HK0.2, there was an improvement in wear resistance (up to 7.5 times) with an increase in hardness. The harder samples showed extensive brittle cracking and surface chipping.

Laser boronizing of maraging steel surfaces using amorphous boron paste has shown high efficiency in terms of increasing wear resistance. Once the problems of softened zones and crack formation have been solved, the method can be applied to improve the wear resistance of typical parts made of maraging steel, for example, the working surfaces of plastics casting molds, whose production volumes of additive technology have been steadily growing recently.

## Figures and Tables

**Figure 1 materials-16-04732-f001:**
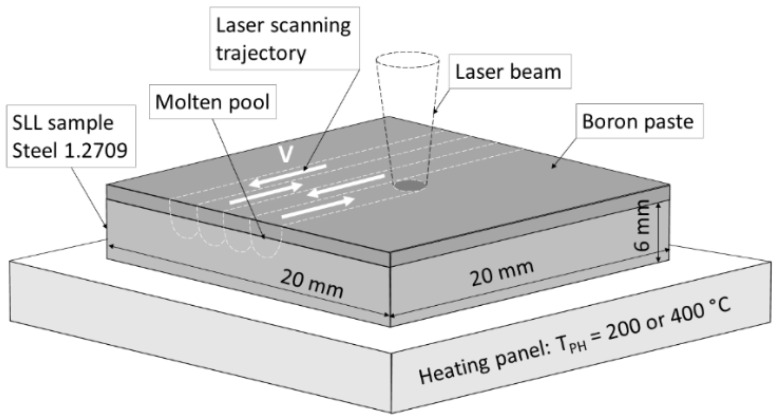
Laser boronizing schema.

**Figure 2 materials-16-04732-f002:**
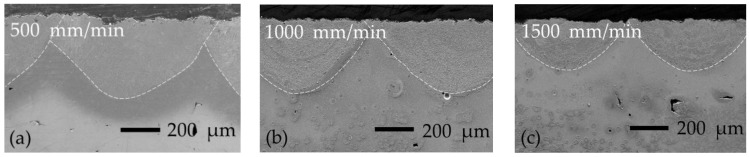
Laser-boronized layers obtained with preheating at 200 °C and 0.03 mm boron paste layer thickness: (**a**–**c**)—500, 1000, and 1500 mm/min laser operating speeds, respectively.

**Figure 3 materials-16-04732-f003:**
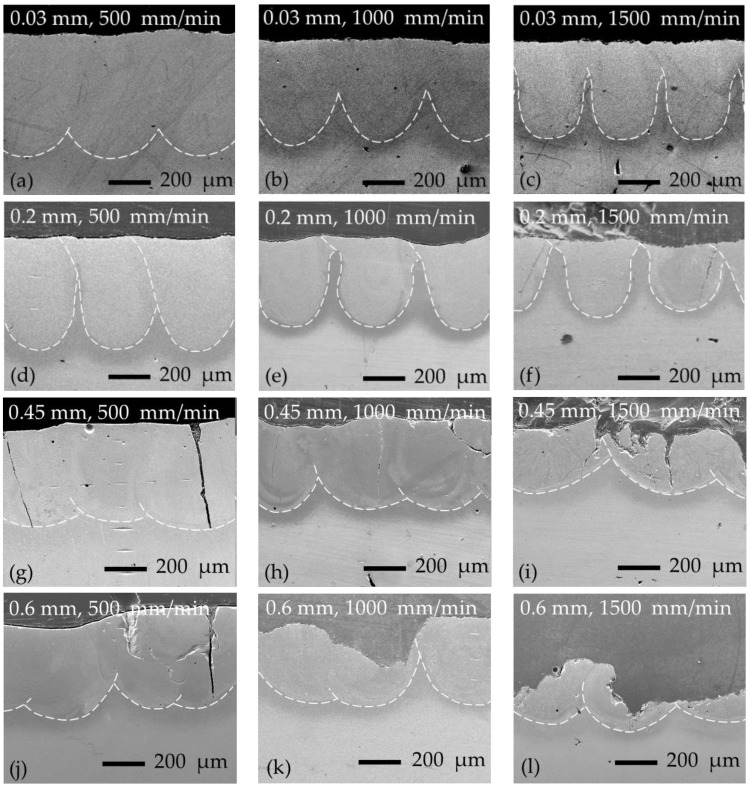
Laser-boronized layers obtained with preheating at 400 °C, 0.03–0.6 mm boron paste layer thickness, and 500–1500 mm/min laser operating speeds: the corresponding boron paste thickness (in mm) and laser operating speed (in mm/min) are indicated in each picture.

**Figure 4 materials-16-04732-f004:**
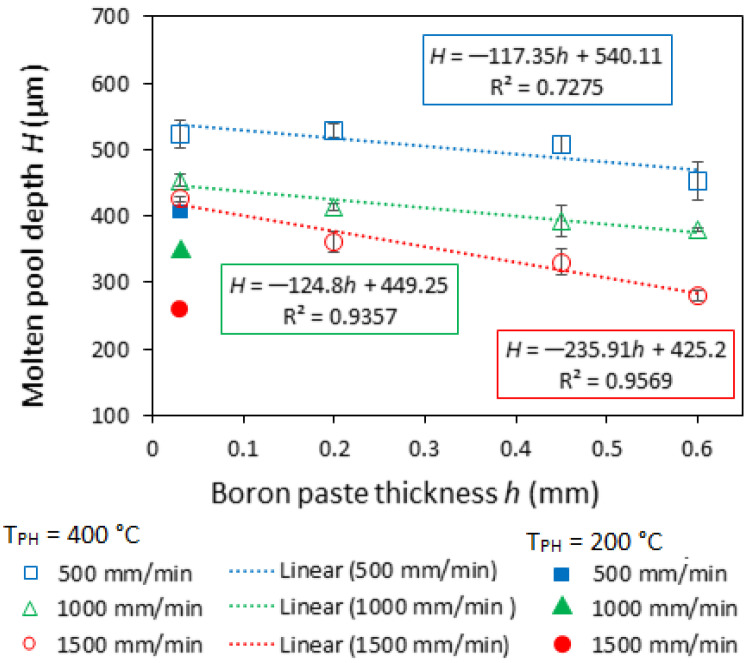
Effect of boron paste thickness and laser operating speed on the molten pool depth.

**Figure 5 materials-16-04732-f005:**
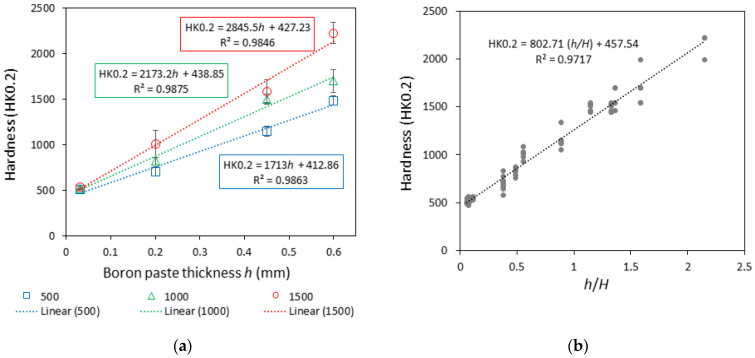
Effect of boron paste thickness and laser operating speed on hardness HK0.2 of laser-boronized layers (**a**) and hardness HK0.2 of laser-boronized layers obtained at different boron paste thicknesses to molten pool depth ratio *h*/*H* (**b**).

**Figure 6 materials-16-04732-f006:**
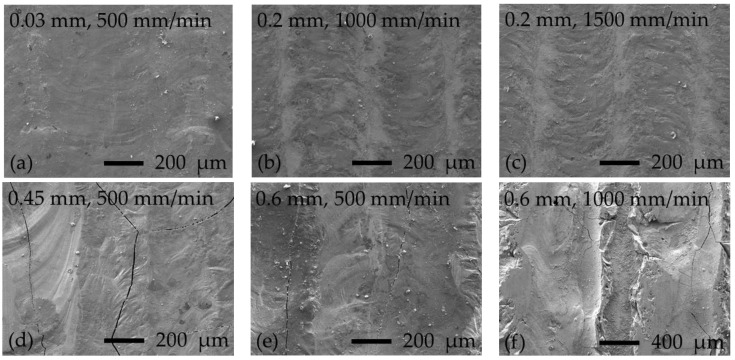
SEM micrographs of laser-boronized surfaces obtained at different boron paste thicknesses and laser operating speeds: the corresponding boron paste thickness (in mm) and laser operating speed (in mm/min) are indicated in each picture.

**Figure 7 materials-16-04732-f007:**
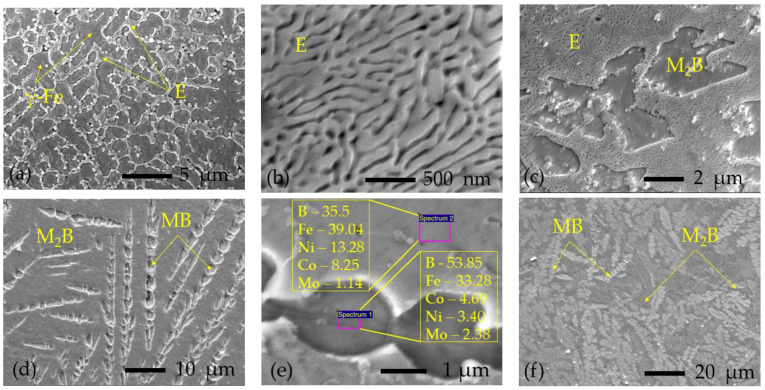
SEM micrographs of typical microstructures observed in laser-boronized layers: (**a**)—hypoeutectic microstructure consisting of primary dendrites of γ-Fe solid solution (γ-Fe) and boride-based eutectic (E) between them; (**b**)—example of fully eutectic microstructure; (**c**)—hypereutectic microstructure of M_2_B borides surrounded by eutectic (E); (**d**)—dendritic higher MB borides in a matrix of lower M_2_B borides; (**e**)—enlarged view of microstructure shown in (**d**) with marked points of X-ray microanalysis and its results in at.%; (**f**)—example of borides’ mixture microstructure with the highest hardness obtained.

**Figure 8 materials-16-04732-f008:**
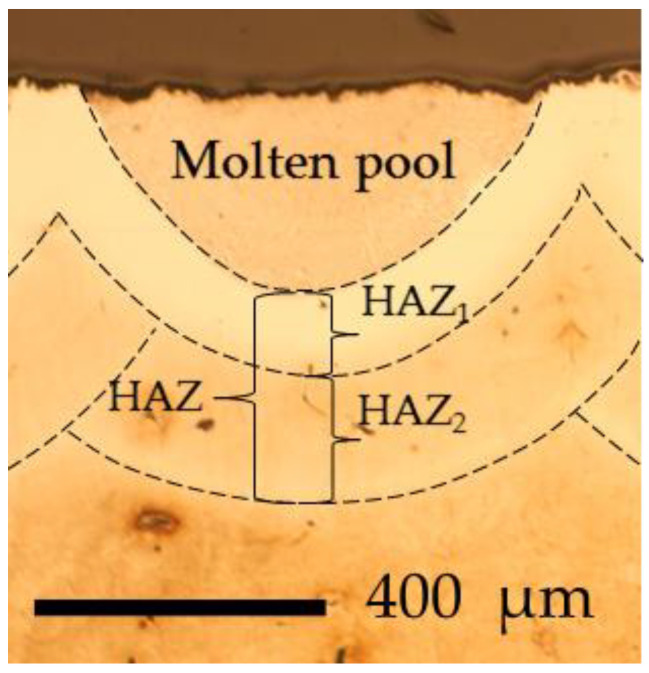
Optical micrograph of cross-sectional view of molten pool with indicated heat-affected zones HAZ_1_, HAZ_2_, and HAZ.

**Figure 9 materials-16-04732-f009:**
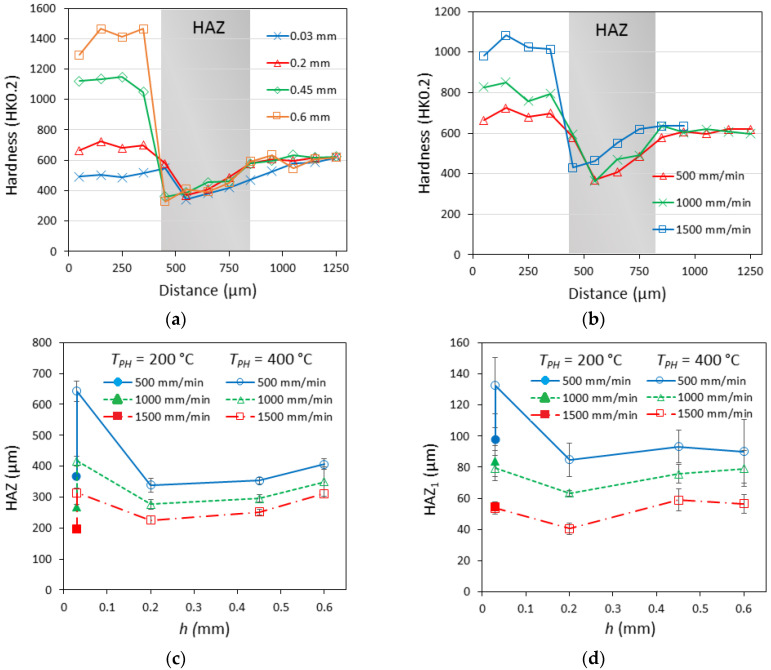
Depth hardness distribution profiles at 500 mm/min laser scanning speed (**a**) and at 0.2 mm boron paste thickness (**b**) variation of heat-affected zones HAZ (**c**) and HAZ_1_ (**d**) at different preheating temperatures, boron paste thicknesses, and laser scanning speeds.

**Figure 10 materials-16-04732-f010:**
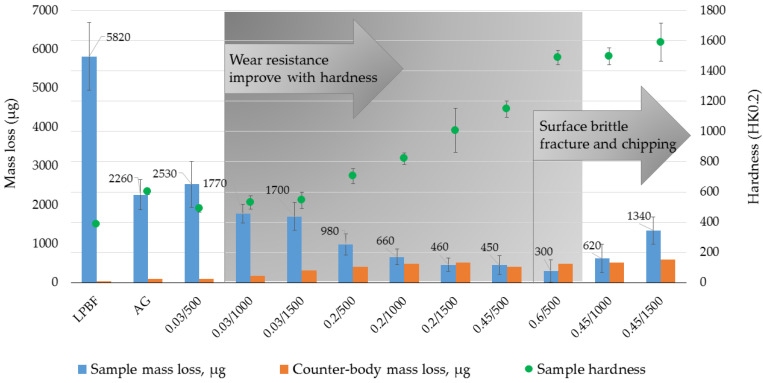
Results of two body dry sliding tests: mass loss of samples manufactured by laser powder bead fusion before (LPBF) and after (AG) aging and boronized at various laser boronizing parameters along with mass loss of counter-body.

**Figure 11 materials-16-04732-f011:**
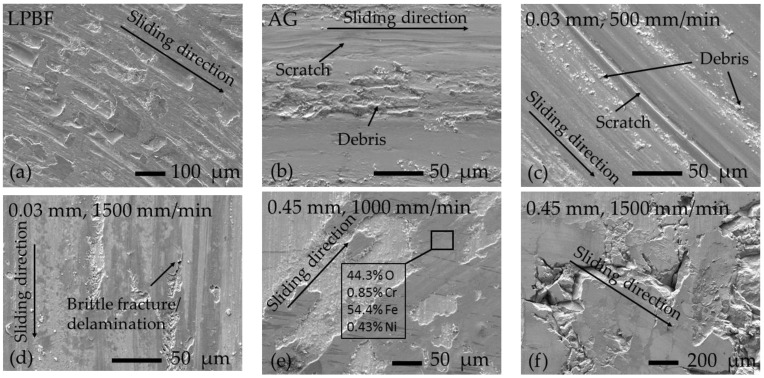
SEM micrographs of wear tracks after two body dry sliding tests: (**a**) control sample manufactured by laser powder bead fusion before (LPBF) without aging; (**b**) LPBF sample after aging (AG); (**c**–**f**) samples boronized at various laser boronizing parameters (boron paste thickness/laser scanning speed); in (**e**)—the elemental composition by X-ray analysis in wt.%.

**Table 1 materials-16-04732-t001:** Summary of experimental results on laser boronizing of metal materials using boron paste [20,21,22,23,24].

Source	Base Material	Laser and Processing Parameters ^1^	Thickness, Hardness of Alloyed Layer	Phase Composition of Alloyed Layer	Notes
[20]	Inconel 718	Ytterbium doped fiber laser, 17.0–28.3 kW/cm^2^, *v_L_* = 3 m/min, boron paste ~250 μm	274–450 µm, ~1350–1610 HV0.5	γ-phase, Ni_2_B, Ni_3_B, CrB, Cr_2_B, FeB, Fe_2_B	Increased roughness and unmelted boron at 17.0–22.6 kW/cm^2^
[21]	Steel 41Cr4	CO_2_ laser, ~37 kW/cm^2^, *v_L_* = 3 m/min, boron paste ~40 μm	~200 µm, 1100–1600 HV	Martensite, Fe_2_B, Fe_3_B	Quick first fatigue crack due to cracks formed during laser processing
[22]	Steel 100CrMnSi6–4	CO_2_ laser, ~37.26 kW/cm^2^, *v_L_* = 2.88 m/min, boron paste ~60 μm	~314 µm, 924–1449 HV	Martensite, borocementit, FeB, Fe_2_B, Fe_3_B	High hardness exceeding 1400 HV obtained only close to the surface
[23]	Steels C20, C45, C90	CO_2_ laser, ~8.28 kW/cm^2^, *v_L_* = 2.88 m/min, amorphous boron paste ~60 μm	~182–239 µm, 867–1037 HV0.05	α-Fe, FeB, Fe_2_B, Fe_3_B	Carbon concentration affects the dilution ratio and hardness
[24]	Steel EN25	CO_2_ laser, 1.5–2.5 kW, *v_L_* = 300–500 mm/min, amorphous boron paste ~250 μm	~540–900 µm, 1150–1315 HV0.05	Mixture of iron borides and martensite	Formation of single passes/No cracks reported

^1^ *v_L_*—laser operating rate; in kW/cm^2^—power density; in kW—laser power.

**Table 2 materials-16-04732-t002:** Parameters of laser boronizing process.

Thickness of Boron Paste Layer *b*, mm	Laser Operating Speed *V*, mm/min		
	500	1000	1500
0.03	*T*_PH_ = 200 °C		
	*s* = 0.7 mm		
0.03	*T*_PH_ = 400 °C		
0.2			
0.45	*s* = 0.4 mm		
0.6			

## Data Availability

Not applicable.

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
