# Peer review of "Fiber Laser Alloying of Additively Manufactured 18Ni-300 Maraging Steel Part Surface: Effect of Processing Parameters on the Formation of Alloyed Surface Layer and Its Properties"

_materials, 2023, doi:10.3390/ma16134732_

Round 1
Reviewer 1 Report
The study presented in this research is sound, and the results produced are interesting. But a revision is required, and after responding to the following remarks and revising the paper, the manuscript may be considered for publication.
1. Literature review needs to include several recent, relevant publications (high impact) highlighting their key findings. The current version only discussed general aspects while the review of each from several papers is necessary. You may provide a review summary table consisting of a column for the comments or key conclusions.
2. More recent relevant literature or similar work discussion is mandatory in the introduction section, which is missing in the Introduction. Authors are suggested to add one paragraph in the introduction section by discussing the recent progress and citing similar work.
3. The novelty of the work is missing in the introduction. Authors are suggested to include a separate paragraph discussing the novelty and importance of the present work.
4. Authors are suggested to include a literature review on the recent publication based on the following references in the introduction section: DOIs: 10.3390/nano12203581; 10.1021/acsaelm.2c00069; 10.1016/j.ceramint.2021.05.167.
Check the typos throughout the manuscript during revision submission also improve the quality of writing.
Reviewer 2 Report
The work is interesting and well described.
My general comment is related to the lacking of some details. In particular I did not find the number of tested samples, for example when the values of hardness are reported (Fig. 8) it is not said if the values in the diagram are mean or single values.
Besides, in the paragraph 3.4 the detected cracks are not specified. Where are the cracks, how many?
Figure 10 is not clear: what do the authors mean with insufficient wear resistance? Which is the criterium?
Specific comments:
pag 3 line 101: The square prism.. should be: square prism
pag. 3 line 119: experiments stage .. should be: experiment stage
pag 4 lines 170-174 the text is not clear, perhaps a table could be more clear
Some figures and some captions are not centred
The quality of english is good
Reviewer 3 Report
This manuscript deals with the strengthening of the working surface of steel parts, obtained by additive manufacturing, by a laser boronising treatment. In particular, the 18Ni-300 MSt specimen surface has been covered by an amorphous boron paste that was processed by a continuous wave fiber laser in melting mode. This technique represents an improvement of the method previously published by the authors, which is based on the use of a continuous CO2 laser. The boronized surface have been characterized by SEM, XRD, Knoop hardness measurements and dry sliding wear tests. According to the obtained results, this laser boronising treatment improved the wear resistance of 18Ni-300 MSt substrate up to 7.5 times.
The manuscript reports about a technique already published by the authors, and the improvement is mainly related to the type of laser used for the surface alloying treatment. Some more detail on the used boron paste and the involved thermochemical process for its transformation in a wear-resistant Fe-B alloy coating should be provided. The specimen cross-section has been investigated at low magnification by scanning electron microscopy (SEM), in Secondary Electron (SE) mode. Why have these flat surfaces not been investigated by classical optical microscopy (i.e., metallography)? Indeed, the colour of the different areas in the cross-section could help significantly in the understanding of the sample morphology. In addition, owing to the different nature of the atoms contained in the coating layer, the SEM investigation could be more conveniently performed in the Backscattered Electron mode (BSE). The formation of macro- and micro-cracks in the very hard and fragile boride phase as a consequence of fatigue stress has not been investigated; authors only discuss about the effect of thermal stress (solidification cracks due to the shrinkage of the brittle layer) on the coating layer. Such information are very important to evaluate the proposed treatment and they should be provided in the revised paper version. Authors should also discuss about the possible ways to solve the coating cracking problem, for example, by creating phases at different boride content (e.g., a tetragonal Fe2B intermediate layer between the steel surface and rhombic FeB layer). English also requires some revisions because a number of misspellings are contained all over the text.
English requires some revisions because a number of misspellings are contained all over the text.
Reviewer 4 Report
The article is devoted to a relevant and interesting topic. Before publishing the article, it would be good to perform a number of improvements.
1. Introduction needs to be improved. There are quite a lot of works devoted to boronizing steels. In the introduction, you should have described in more detail the difference between boron plating of conventional forged blanks and boron plating of blanks obtained by additive technologies. Also in the introduction you have practically no analogues of your work on the topic of boronizing . That is, more work on boronizing should be described. Write what structure was obtained, what were the problems.
2. Samples after their manufacture by additive technologies are often subjected to mechanical processing. In order to obtain the required surface quality. This question is not covered in your article. It would be necessary to add which blanks should be borated (before or after machining). Also write what surface quality will be after borating and whether it is possible to process the workpiece after borating.
3. In the article itself, you also do not give results on the effect of this treatment on the quality of the surface. At least some data should have been provided. After all, this is extremely important for practical use.
4. The procedure should indicate what quality of the surface should be for the boriding operation. Indeed, after additive technologies, a very high surface roughness can be obtained.
5. The work should dwell in more detail on the possibility of practical application of the results of the work. After all, in general, the work is focused on solving a production problem. Supplement the conclusions with the possibility of practical application of the results of the work.
6. In the introduction, you talk about the similarity of the properties of blanks obtained by additive technologies and forged blanks. In the work, it would be useful to compare the results obtained by you and similar data for forged blanks. Show differences and similarities of results. For example, to compare with the results of other authors published works on boriding forged blanks from similar materials.
Reviewer 5 Report
The author presents an experimental study on the increase in resistance and hardness of a maraging steel after application of boron paste layer with application by fiber laser.
The text has few typos; I present some in the attachment.
The manuscript is clearly just an application of laser powder processing and surface treatment, namely laser alloying.
The work is well understandable and credible; the techniques used seem to me to be adequate; however, this is just another article similar to many others and I don't think it brings anything especially new. I suggest the author focus more on that part.

Round 2
Reviewer 2 Report
The manuscript is now acceptable
Reviewer 3 Report
The revised version of the manuscript has been deeply modified/improved, according to all amendments suggested by the reviewer. This revised manuscript version can be accepted for publication on the journal Materials in its present form.
Reviewer 4 Report
The authors carefully and well finalized the article. The article can be published in my opinion.